# Adaptive Phase Transform Method for Pipeline Leakage Detection

**DOI:** 10.3390/s19020310

**Published:** 2019-01-14

**Authors:** Yifan Ma, Yan Gao, Xiwang Cui, Michael J. Brennan, Fabricio C.L. Almeida, Jun Yang

**Affiliations:** 1Key Laboratory of Noise and Vibration Research, Institute of Acoustics, Chinese Academy of Sciences, Beijing 100190, China; shanximayifan@163.com (Y.M.); cuixiwang2010@126.com (X.C.); jyang@mail.ioa.ac.cn (J.Y.); 2University of Chinese Academy of Sciences, Beijing 100049, China; 3Department of Mechanical Engineering, UNESP, Ilha Solteira, SP 15385-000, Brazil; mjbrennan0@btinternet.com; 4Faculty of Science and Engineering, UNESP, Tupã, SP 17602-496, Brazil; fabricio.lobato@yahoo.com.br

**Keywords:** LMS adaptive algorithm, phase spectrum, pipeline leakage detection, time delay estimation

## Abstract

In leak noise correlation surveys, time delay estimation (TDE) is of great importance in pinpointing a suspected leak. Conventional TDE methods involve pre-filtering processes prior to performing cross-correlation, based on a priori knowledge of the leak and background noise spectra, to achieve the desired performance. Despite advances in recent decades, they have not proven to be capable of tracking changes in sensor signals as yet. This paper presents an adaptive phase transform method based on least mean square (LMS) algorithms for the determination of the leak location to overcome this limitation. Simulation results on plastic water pipes show that, compared to the conventional LMS method, the proposed adaptive method is more robust to a low signal-to-noise ratio. To further verify the effectiveness of the proposed adaptive method, an analysis is carried out on field tests of real networks. Moreover, it has been shown that by using the actual measured data, improved performance of the proposed method for pipeline leakage detection is achieved. Hence, this paper presents a promising method, which has the advantages of simple implementation and ability to track changes in practice, as an alternative technique to the existing correlation-based leak detection methods.

## 1. Introduction

Urban pipelines are usually seen as an environmentally friendly means for transporting fluids and gases [1]. Buried pipelines are badly deteriorated, leading to leakage and burst as a result of aging water supply facilities, poor management, and inappropriate excavation practices [2,3]. More than 32 billion cubic meters of water is lost within the distribution systems, which accounts for 35% of the total water supply across the world [4]. Moreover, as the largest developing country in the world, China has been challenged by its water scarcity situation to maintain the balance between economic growth and rapid urbanization [5]. The gap between the available water supply and increasing demand is exacerbated due to water loss; this, in turn, has profound impacts on social, economic, and environmental wellbeing [6]. There is a growing awareness within the water industry and leak detection community that more effective pipe leak detection technologies are needed in their attempts to fulfill high demands of reliability and integrity of pipelines. Further push toward more efficient real-time monitoring systems for automated leakage detection in buried pipelines is leading to major changes associated with water resources and water quality management.

Great efforts have been made to develop leak detection methods, which are mainly divided into hardware- and software-based methods [7,8,9,10,11,12,13,14,15,16,17,18,19,20,21,22,23,24,25,26,27,28,29,30,31,32,33]. When it comes to locating a leak with hardware-based methods, optical fibers and cable sensors can be adopted. In contrast, software-based methods primarily require signals captured by sensors to provide sufficient information about the leak source, such as negative pressure waves and fluid waves. Among these techniques, cross-correlation methods where leak noise is measured at either side of a suspect leak have proved to be effective, and have been widely used to locate leaks in buried water pipelines for many years [31,32,33]. Previous work has shown that the leak noise is concentrated at low frequencies (typically up to 1 kHz for metal pipes and 200 Hz for plastic pipes), with the dominant waveform being the *s* = 1 wave as the main carrier of vibrational energy induced by water leakage. Moreover, the speed of leak noise can be seen as a constant, with the effect of surrounding soil at low frequencies being negligible [34]. Currently available cross-correlation leak detection systems operate on passive means to estimate the difference in arrival times between the pertinent acoustic waves measured by two sensors. A wide spectrum involving cross-correlation for time delay estimation (TDE) was first discussed by Knapp and Carter [33] and their performance for water leak detection was further discussed by Gao et al. [32]. More recently Gao et al. [35] have investigated the implementation of differentiation process (termed DIF) in the cross-correlation analysis. They showed that the DIF method has the ability to eliminate some of the ambiguity caused by interference at low frequencies, and thus enables a more pronounced and reliable peak of cross-correlation corresponding to the true time delay resulting from the propagating leak noise. On the other hand, Piersol [36] showed that the time delay can be estimated by using phase data with no penalty in the form of reduction of prediction accuracy. Zhao and Hou [37] extended his work and set up a theoretical frame of the generalised phase spectrum method by using a frequency-dependent weighting function. Brennan et al. [38] further validated the equivalence of the cross-correlation methods and generalised phase spectrum methods (i.e., the duality relationship between the time and frequency domains).

It should be noted that statistical characteristics of both leak and background noise signals cannot be known precisely in various practical situations, which may hinder the effectiveness of correlation methods by passive means. In response to this, the least mean square (LMS) algorithm provides a solution for TDE. This has an apparent advantage: It can extract adaptively the similarity between two sensor signals without considering their statistical characteristics, based on the minimum mean square error. However, the conventional LMS algorithm is based on the assumption that the signals received by two sensors only have the time delay and attenuation in amplitude (being frequency independent). This can limit the application of the LMS method for leak detection in pipelines because of the dispersion behaviour of the leak noise.

Based on the propagation model of leak signals in buried water pipes, an adaptive phase transform (ADPHAT) method is proposed in this paper. Two impulse response functions (IRFs) are first obtained by using the LMS algorithm for two sensor signals, and are subsequently transformed into the frequency domain to result in a new frequency response function (FRF) that only contains the phase information directly related to the time delay. Simulation and field tests are conducted to evaluate the performance of the proposed method in comparison with the conventional LMS algorithms for TDE. To assist the reader, the process of the LMS adaptive algorithm is briefly discussed in Appendix A. Notations are explained in Nomenclature.

## 2. Methodology

### 2.1. Water Leak Detection

Basic principles of water leak detection are briefly discussed in this section to facilitate the understanding of the process of the proposed adaptive algorithm. When leakage occurs in a pipe system, the dominant propagating wave at low frequencies is the axisymmetric fluid-borne wave with a wavenumber approximated by [39]
(1)$$k^{2}=k_{f}^{2}\left(1+\frac{2Ba}{Eh+i\eta Eh}\right)$$
where *k_f_* is the free-field fluid wavenumber; *η* is the loss factor of the pipe wall; *a* and *h* are the pipe radius and wall thickness respectively; *E* is the Young’s modulus of the pipe wall material; and *B* is the fluid bulk modulus of elasticity. The real and imaginary parts of the wavenumber result in the propagation speed and wave attenuation of the leak noise. From Equation (1), the propagation speed and attenuation factor of leak noise can be obtained by
(2a)$$c=c_{f}{\left(1+\frac{2Ba}{Eh}\right)}^{-1/2}$$
and
(2b)$$\beta =\frac{1}{c_{f}}\frac{\eta Ba/Eh}{{\left(1+2Ba/Eh\right)}^{1/2}}$$
Equation (2) shows that leak noise is predominantly non-dispersive with a constant attenuation factor *β*.

Referring to Figure 1, for leak detection in water distribution pipes, sensors such as accelerometers or hydrophones are generally deployed at two access points (e.g., pipe sections, hydrants, or valves) in order to measure the vibration or acoustic signals generated by a suspected leak.

The leak location (relative to sensor 1), *d*_1_, can be determined using the basic algebraic relationship given by
(3)$$d_{1}=\frac{D+cT_{0}}{2}$$
where *D* is the distance between two sensors and *T*_0_ is the estimate of the time delay between two measured signals. Equation (3) shows that accurate estimation of the time delay plays a dominant role in water leak detection. Various approaches to TDE have been proposed and implemented, but it is believed that for this particular application, the correlation methods yield optimal time delay estimators that correspond to distinct peaks in the cross-correlation functions, if a leak exists [33].

Noise from a leak, *l*(*t*), is measured by two sensors, *x*_1_(*t*) and *x*_2_(*t*), as illustrated in Figure 2. Here, *H*(*ω*,*d*_1_) and *H*(*ω*,*d*_2_) denote the FRFs between the leak and sensors 1 and 2 respectively; *n*_1_(*t*) and *n*_2_(*t*) are the background noise signals that are assumed to be random Gaussian, and uncorrelated with each other and with the leak signal. The sensor signals, *x*_1_(*t*) and *x*_2_(*t*), can thus be modelled by
(4a)$$x_{1}(t)=h_{1}(t)⊗l(t)+n_{1}(t)$$
(4b)$$x_{2}(t)=h_{2}(t)⊗l(t)+n_{2}(t)$$
where ⊗ denotes convolution; *h*_1_(*t*) and *h*_2_(*t*) are the IRFs of the sensor signals 1 and 2 relative to the leak source respectively, and given by $h_{1}(t)=F^{-1}\{H(\omega ,d_{1})\}$ and $h_{2}(t)=F^{-1}\{H(\omega ,d_{2})\}$; $F^{-1}$ denotes the inverse Fourier transform (FT). Correspondingly the sensor signals are seen as the leak noise *l*(*t*) passed through the pipe system in the presence of background noise.

Combining Equations (4a) and (4b), the cross-spectral density (CSD) between the two sensor signals, *x*_1_(*t*) and *x*_2_(*t*), can be obtained by
(5)$$S_{12}(\omega )=S_{ll}(\omega )H^{*}(\omega ,d_{1})H(\omega ,d_{2})+\tilde{S}(\omega )$$
where * denotes complex conjugation; *S_ll_*(*ω*) is the auto-spectral density (ASD) of the leak noise, *l*(*t*); and the remaindering term $\tilde{S}(\omega )$ is attributable to the background noise, which is generally trivial in comparison with the CSD of the leak signals. Neglecting the term $\tilde{S}(\omega )$, Equation (5) is further expressed by
(6)$$S_{12}(\omega )=S_{ll}(\omega )e^{-\omega \beta D+j\omega T_{0}}$$
with the phase obtained by
(7)$$\Phi (\omega )=\mathrm{Arg}\{S_{12}(\omega )\}=\omega T_{0}$$

It is clear from Equation (6) that the phase of the CSD (i.e., phase spectrum) between the sensor signals is related directly to the time shift experienced by the leak signals as they propagate along the pipe. Indeed, calculation of the phase gradient of the CSD with respect to frequency leads to an estimate of the time delay [33,38].

### 2.2. The Proposed ADPHAT Method

The process of the conventional LMS adaptive algorithm is given in detail in Appendix A. It is shown that in the mathematical model of two sensor signals with a pure time delay, the LMS algorithm may lead to an accurate estimate of the time delay *T*_0_. Of particular interest in this paper, however, is the decaying leak noise with the attenuation factor *β*. In this circumstance, a large error in the detection of a leak location may occur if the conventional LMS adaptive method is adopted, thus restricting its application in practical leak detection surveys. A strong motivation for expanding on the conventional LMS algorithm is its simple implementation, and its ability to track changes in sensor signals without any concern with the spectral information about the leak noise and background noise signals. Another disadvantage of the conventional LMS adaptive algorithm is that in the process, calculation of the phase gradient of the IRF introduces a reference time delay, *T*_ref_. This is addressed explicitly as follows.

For the conventional LMS algorithm as discussed in Appendix A, an IRF, *h*(*n*), is generated. Here *n* is an integer, *n* = −*P*~*P*, and *P* is the maximum order in the LMS algorithm calculated by the maximum time delay. FT of *h*(*n*) gives
(8)$$F\{h(n)\}=\sum_{n=-P}^{P}h(n)e^{-j\omega (n+P)/fs}$$
where F denotes the FT. For two sensor signals, there are two possible IRFs, *h*_12_(*n*) and *h*_21_(*n*), obtained from *x*_1_(*t*) and *x*_2_(*t*) being the input and expected signals and vice versa. Note that if *h*_12_(*n*) has the maximum values at *n* = *N*, then the maximum value of *h*_21_(*n*) occurs when *n* = −*N*. In general, a sharp peak in the IRF is desirable for TDE. Correspondingly, neglecting the contributions of the rest terms in Equation (8) is an approximate approach to the FRF between the two sensor signals. This leads to
(9)$$H(\omega )\approx {F\{h(n)\}|}_{n=N}=h(N)e^{-j\omega (N+P)/fs}$$
Consider the following two cases:(1)If there is no time delay between two signals *x*_1_(*t*) and *x*_2_(*t*) (i.e., *T*_0_ = 0), then the maximum value of the IRF *h*(*n*) occurs at *n* = 0. Setting *n* = 0, Equation (9) leads to the two FRFs given by
(10a)$$H_{12}(\omega )\approx {F\{h_{12}(n)\}|}_{n=0}=h_{12}(0)e^{-j\omega P/fs}$$
(10b)$$H_{21}(\omega )\approx {F\{h_{21}(n)\}|}_{n=0}=h_{21}(0)e^{-j\omega P/fs}$$
It is clear from Equation (10) that when *T*_0_ = 0, the gradient of the phase spectrum is −*P*/*fs*, which is defined as the reference time delay, *T*_ref_, by
(11)$$T_{\mathrm{ref}}=-P/fs$$(2)When the time delay $T_{0}\neq 0$, FTs of the two IRFs lead to
(12a)$$H_{12}(\omega )\approx {F\{h_{12}(n)\}|}_{n=N}=h_{12}(N)e^{-j\omega (P+N)/fs}$$
(12b)$$H_{21}(\omega )\approx {F\{h_{21}(n)\}|}_{n=-N}=h_{21}(-N)e^{-j\omega (P-N)/fs}$$
From Equations (12a) and (12b), the phase spectra between two sensor signals can be obtained by
(13a)$$\Phi_{12}(\omega )=\mathrm{Arg}\{H_{12}(\omega )\}=-\omega (P+N)/fs$$
(13b)$$\Phi_{21}(\omega )=\mathrm{Arg}\{H_{21}(\omega )\}=-\omega (P-N)/fs$$

Equation (12) shows that the phase of *H*_12_(*ω*) and *H*_21_(*ω*) contains both the time delay, *T*_0_, and the reference time delay, *T*_ref_. Correspondingly, the FRFs, *H*_12_(*ω*) and *H*_21_(*ω*), by removing *T*_ref_, give rise to the information of the time delay *T*_0_ = *N/fs*. An alternative FRF is proposed in order to eliminate the reference time delay in the resultant phase information, and given by
(14)$$H(\omega )=\sqrt{H_{12}(\omega )H_{21}^{*}(\omega )}\approx \sqrt{h_{12}(N)h_{21}(-N)]e^{-j\omega N/fs}}$$

It can be seen from Equation (14) that despite the unknown spectral characteristics of leak source, the proposed FRF, *H*(*ω*), only contains the phase information related to the time delay that can be achieved directly by calculating the gradient of the phase spectrum.

### 2.3. Improvement in TDE

In the development of the new FRF, it is important to note that ideally, a distinct peak can be identified in each IRF produced by using the conventional LMS adaptive algorithm (see Appendix A). This a good approximation of the FRF between the two sensor signals to be determined by considering only the respective peak values of *h*_12_(*n*) and *h*_21_(*n*). In reality, however, the contributions of the remaining terms in Equation (8) may be relatively large compared to the peak values. To enhance the performance of the proposed TDE algorithm, an additional procedure is conducted prior to the FTs of the IRFs. By taking the square of each IRF, the corresponding peak values can be significantly accentuated. The proposed ADPHAT algorithm for TDE can be obtained from the inverse FT of the new FRF, and is given by
(15)$$h(n)=F^{-1}\left\{\sqrt{H_{12}(\omega )H_{21}^{*}(\omega )}\right\}$$

A procedure for the implementation of the ADPHAT method for TDE is illustrated in Figure 3. In the reminder of this paper, the simulated and actual leak signals will be adopted to evaluate the effectiveness of the proposed method.

## 3. Numerical Examples

In this section, numerical examples of plastic water pipes are presented to verify the performance of the ADPHAT method proposed in this paper for TDE. In the simulation, both the leak source and background noise signals are white Gaussian with zero means. To emulate the pipe system as illustrated in Figure 2, the parameters of the plastic pipe are set as *β* = 2 × 10^−4^ s/m, *d*_1_ = 22.8 m, *d*_2_ = 38 m, *c* = 380 m/s. The arrival times of the leak source and the two sensors are calculated to be *t*_1_ = 0.06 s and *t*_2_ = 0.1 s respectively, which gives the actual time delay *T*_0_ = −0.04 s. The simulated leak source signal then propagates downstream and upstream of the pipe system, and is captured at a sampling frequency of *fs* = 1 kHz. Consider the low signal-to-noise ratio (SNR) case, for example SNR = −10 dB. The ASDs of the resultant sensor signals at *d*_1_ and *d*_2_ are shown in Figure 4. As can be seen in the figure, the ASDs decrease dramatically with increasing frequency and drop to the level of background noise below 50 Hz, suggesting the simulated signals are concentrated at lower frequencies as anticipated. In the analysis, the possible maximum number for the signals generated is set to be *P* = 100 in the LMS filter coefficients, as suggested in the appendix. This further yields a reference time delay *T*_ref_ = −*P*/*fs* = −0.1 s, as defined by Equation (11). In the conventional LMS adaptive method, *h*_12_(*n*) denotes the IRF for the input signal *x*_1_(*t*) and the expected signal *x*_2_(*t*); *h*_21_(*n*) denotes the IRF for the input signal *x*_2_(*t*) and the expected signal *x*_1_(*t*). It must be borne in mind that the reference time delay *T*_ref_ needs to be removed in the estimates of the time delay given by these two algorithms to ensure accurate TDE.

Figure 5 plots the IRFs, *h*_12_(*n*) and *h*_21_(*n*), using the conventional LMS adaptive algorithms. The time delay estimates are found to be *T*_0_ = −0.039 s and 0.041 s for *h*_12_(*n*) and *h*_21_(*n*) respectively. However, the peak values, as shown in the figure, are less convincing, since they are barely distinguishable in the IRFs due to some fluctuations caused by background noise. This is, indeed, confirmed when observing the corresponding FRFs in the frequency domain as plotted in Figure 6, by virtue of the fact that the non-dispersive behaviour is dominant in the narrow bandwidth below 50 Hz (i.e., the phase can clearly be seen to vary linearly with frequency). Once the two IRFs have been determined, they can be adopted to lead to the new FRF *H*(*ω*) by following the procedures as illustrated in Figure 3. The phase of the resulting *H*(*ω*) is shown in Figure 6 in comparison with those given by the conventional LMS adaptive method. The evidence is compelling: The phase of *H*(*ω*) has a linear relationship for the frequency range up to 400 Hz, which dramatically broadens the bandwidth over which the TDE can be conducted when subjected to the deteriorating effect of background noise. The IRF is shown in Figure 7, from which a more accurate time delay estimate of *T*_0_ = −0.04 s can be obtained, corresponding to the peak value in the proposed IRF, *h*(*n*). Moreover, compared to the results given by the conventional LMS adaptive method as plotted in Figure 5, it can be seen that the ADPHAT method gives a more pronounced peak with fewer fluctuations in the time domain representation.

The performance of the LMS adaptive methods for TDE is shown in Figure 8 as a function of SNR. In general, it can be observed that the performance of each adaptive algorithm provides consistency in TDE. As can be seen in Figure 8b, the conventional LMS algorithms, *h*_12_(*n*) and *h*_21_(*n*), lead to errors of 1%~3% in TDE for different SNRs, whereas the new IRF, *h*(*n*), with relative error less than 1%, outperforms the IRFs given by the conventional LMS methods. In particular, when the SNR is very poor (SNR = −20 dB), the proposed IRF, *h*(*n*), can still achieve very accurate time delay estimate. The reason *h*(*n*) gives the best estimate of the time delay is that the leak noise signals are both corrupted by background noise at locations *d*_1_ and *d*_2_, which can, to some extent, be compensated for by multiplying the two FRFs, *H*_12_(*ω*) or *H*_21_(*ω*), prior to the FT. Furthermore, it must be pointed out that in practice the leak location is unknown, and thus *h*(*n*) will also provide an optimal solution for TDE based on the calculated *h*_12_(*n*) and *h*_21_(*n*).

## 4. Experimental Work

The leak detection data presented in this section were captured from a live water distribution system. Filed tests were carried out at a leak detection facility connected to the mains located at the campus of the Institute of Acoustics, Chinese Academy of Sciences (IACAS), a schematic map of which is shown in Figure 9. The pipeline consisted of 200 mm OD cast iron, and was buried at a depth of approximately 2 m. With reference to Figure 9, the testing was conducted on a straight pipe section of 100 m with three accelerometers attached magnetically to the pipe wall through three access manholes shown in Figure 10. As illustrated in Figure 11, by opening a valve, a leak from a pipe section was generated above the ground at a distance of 6.2 m from access point 2.

The leak signals were captured by using a B&K 3050-B-060 multi-channel noise analyser. The sampling frequency of 8192 Hz was applied to the accelerometer-measured signals for a duration of 10 s (that may give a discrepancy in the TDE due to a time domain resolution of approximately 10^−4^ s). The parameters of cast iron were *c_f_* = 1500 m/s, *B* = 2.25 GPa, and *E* = 140 GPa. Substituting these pipe system parameters into Equation (2a), the propagation speed of the leak signal was calculated as *c* = 1333 m/s. Leak detection measurements were made by using accelerometer pairs at access points 2 and 3 and access points 1 and 3 (denoted as A2&A3 and A1&A3 as plotted in Figure 10) respectively. In the following subsections, the adaptive methods are applied to these two pairs of accelerometer-measured signals that bracketed the leak.

### 4.1. Case One 

For the accelerometer pair located at A2&A3, the distances of the two accelerometers were 6.2 m and 43.8 m, relative to the leak, respectively. From Equation (3), a true time delay of 0.0282 s can be calculated. The time series of the two accelerometer-measured signals are plotted in Figure 12. The signals were then processed via a 1024-point FFT using a Hanning window with 50% overlap. Figure 13a,b show the frequency results, including the coherence function and the phase spectrum, respectively, between the accelerometer pair. It can be seen from Figure 13a that the signals measured by the accelerometers are concentrated mainly in the frequency range between 100 Hz and 3200 Hz (except between 1 kHz and 1400 Hz) where a good coherence is found, otherwise the coherence is very low since background noise dominates.

The time delay estimate using the proposed ADPHAT method is now made in comparison with those given by conventional LMS adaptive algorithms. Using the empirical formula defined in Appendix A, the order of the IRF can be taken as *P* = 250, which results in a reference time delay calculated as *T*_ref_ = −*P*/*fs*= −0.0305 s. Figure 14 shows the steady IRFs (after 2 × 10^5^ iterations) using the conventional LMS algorithms. Here, *h*_12_ denotes the result for the input signal at A2 and the expected signal at A3; *h*_21_ denotes the result for the input signal at A3 and the expected signal at A2. The maximum values in the figure lead to time delay estimates of 0.0282 s and −0.0283 s for *h*_12_ and *h*_21_ respectively. Figure 15 plots the new FRF *H*(*ω*) achieved in the implementation of the ADPAHT method, which highlights a linear behaviour of the phase in the entire frequency range of interest. By taking the inverse FT of the FRF, a distinct peak can be obtained in the IRF as plotted in Figure 16, which corresponds to the time delay estimate of *T*_0_ = 0.0282 s. Compared to the results given by the conventional LMS adaptive method, a more distinguishable time delay estimate can be obtained by the proposed ADPHAT method.

### 4.2. Case Two

Consider the accelerometer pair located at A1&A3, the distances of which were 35.0 m and 43.8 m relative to the leak source. A true time delay of 0.0066 s can be calculated from Equation (3). Figure 17 plots the original time domain signals captured by A1&A3. Similar to the frequency analysis performed on A2&A3, the results of the coherence and phase spectrum are shown in Figure 18. It can be seen from Figure 18a that the two signals have good coherence in the frequency 100~3200 Hz, where the phase does not vary linearly with frequency. In light of this, inaccuracies in the time delay estimates by directly adopting the conventional LMS adaptive method is perhaps not surprising. Nevertheless, by using the proposed method, an approximately linear relationship is found in the new FRF, *H*(*ω*), in the entire frequency range of interest, as shown in Figure 19. This further leads to the IRF as plotted in Figure 20. A sharp peak is found in the IRF corresponding to an accurate time delay estimate of *T*_0_ = 0.0066 s.

## 5. Conclusions

In this paper, an ADPHAT method for TDE has been developed for pipeline leakage detection. It has expanded on the conventional LMS algorithm, and developed a new way to calculate the time delay of the sensor signals. The leakage propagation characteristics of pipelines have been accounted for in the implementation process of the proposed adaptive algorithm, which results in an FRF containing only the phase information of two leak signals. Due to the duality of both the time and frequency domains, the time delay is estimated by searching the maximum value of the inverse FT of the new FRF. Simulations on plastic water pipes demonstrate the effectiveness of the proposed method for locating leaks even in a low SNR environment. Field tests were further conducted to confirm the improved performance compared to the conventional LMS algorithms for TDE, which is particularly well-suited to pipeline leakage detection. An additional advantage of this approach over the existing correlation-based leak detection methods is that it does not rely on the selection of pre-filtering/frequency weighting of input signals, and potentially could interpret the similarity between two sensor signals in a low SNR.

## Figures and Tables

**Figure 1 sensors-19-00310-f001:**
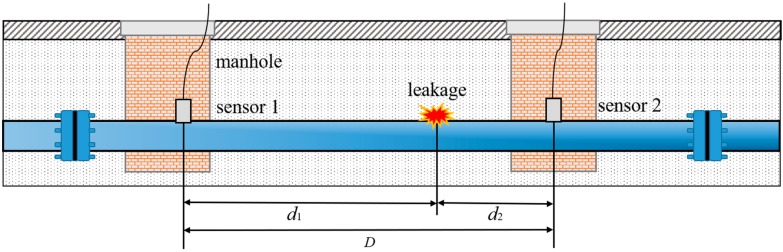
Schematic of leak detection measurements in water distribution pipes.

**Figure 2 sensors-19-00310-f002:**
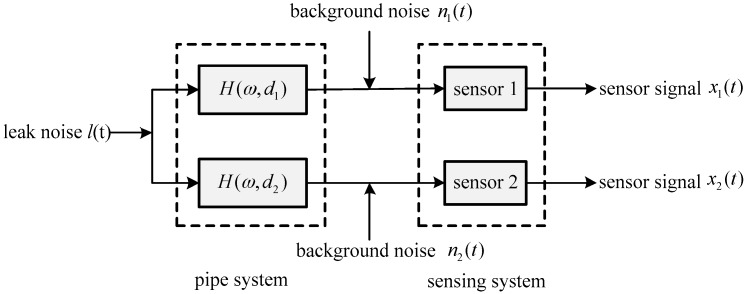
Flow diagram of leak noise measured by two sensors.

**Figure 3 sensors-19-00310-f003:**
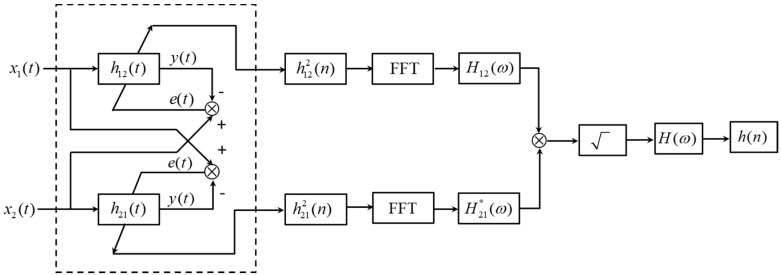
Implementation of the ADPHAT method.

**Figure 4 sensors-19-00310-f004:**
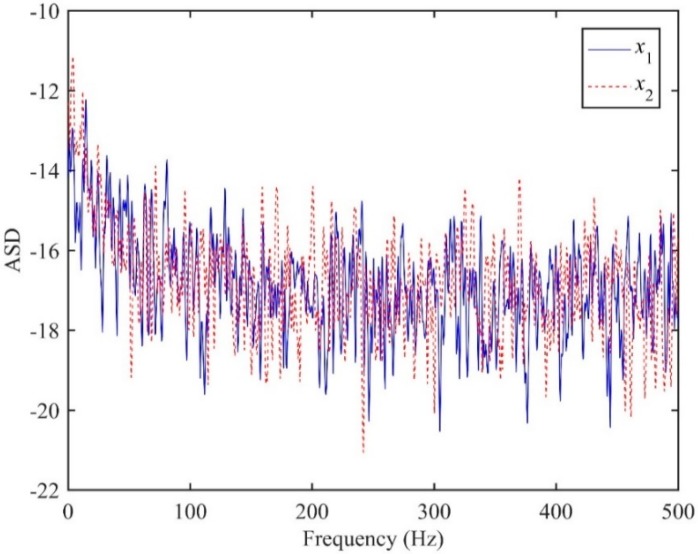
ASDs of the two sensor signals when SNR = −10.

**Figure 5 sensors-19-00310-f005:**
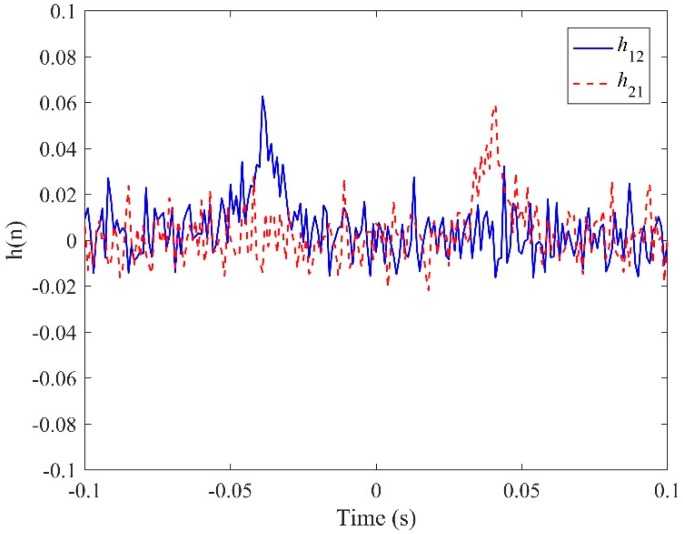
IRFs given by the conventional LMS adaptive method.

**Figure 6 sensors-19-00310-f006:**
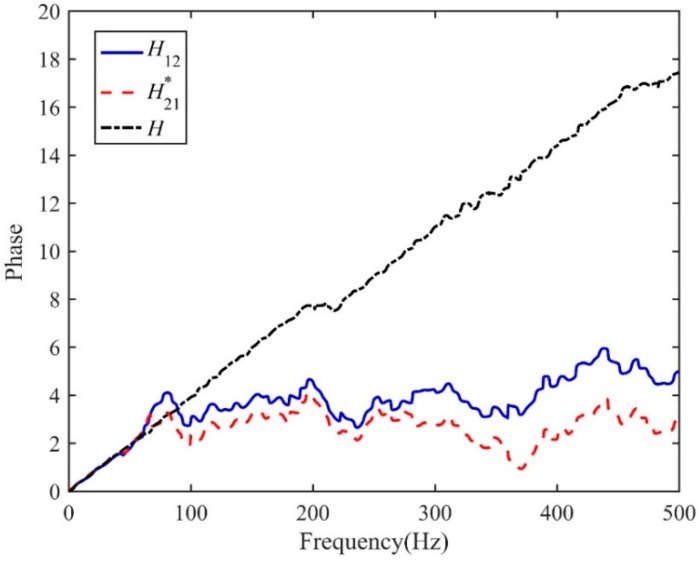
Phase of the FRF.

**Figure 7 sensors-19-00310-f007:**
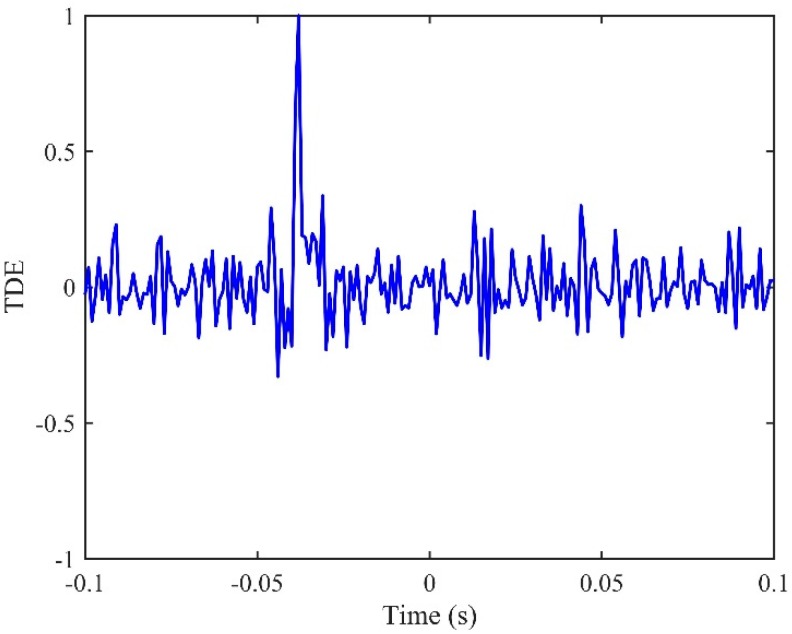
The IRF given by the proposed ADPHAT method for TDE.

**Figure 8 sensors-19-00310-f008:**
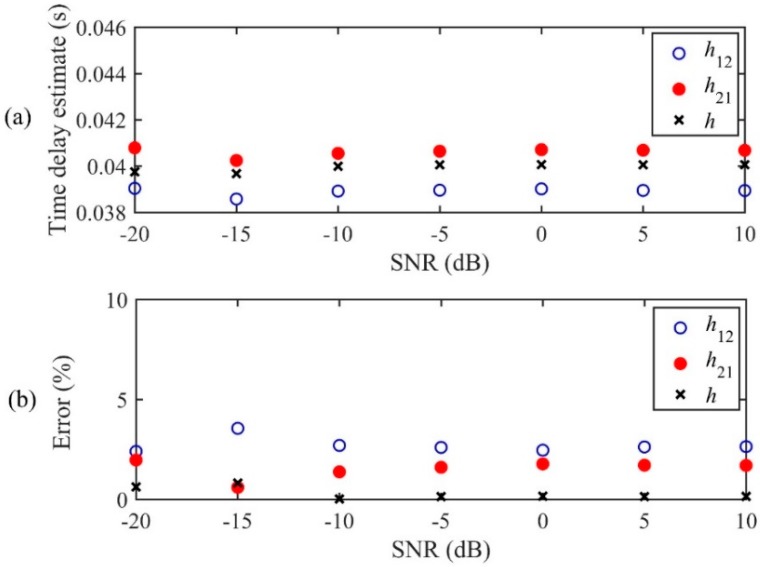
Effect of background noise on the performance of the adaptive methods for TDE: (**a**) Time delay estimate; (**b**) relative error.

**Figure 9 sensors-19-00310-f009:**
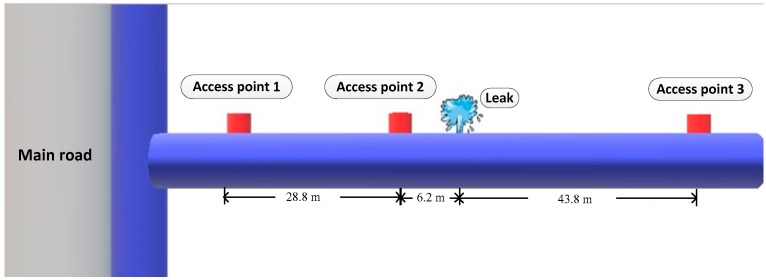
Schematic of the test site located at the IACAS.

**Figure 10 sensors-19-00310-f010:**
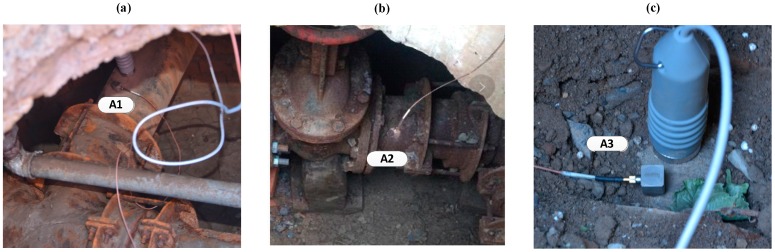
Accelerometers attached magnetically to the test pipe. (**a**) At access point 1; (**b**) At access point 2; (**c**) At access point 3.

**Figure 11 sensors-19-00310-f011:**
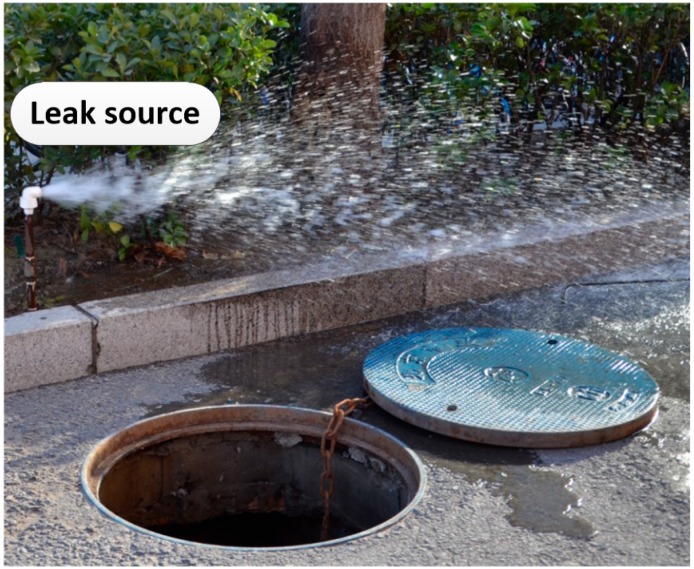
Leak generated above the ground.

**Figure 12 sensors-19-00310-f012:**
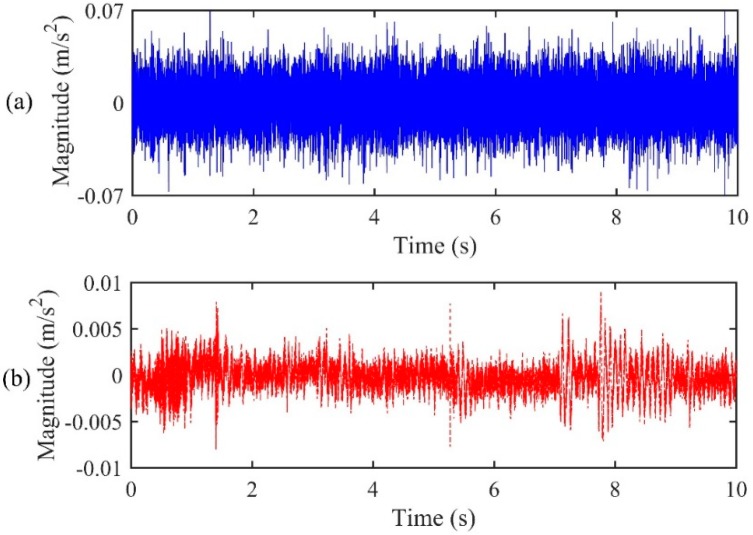
Leak signals measured by the accelerometer pair of (**a**) A2; (**b**) A3.

**Figure 13 sensors-19-00310-f013:**
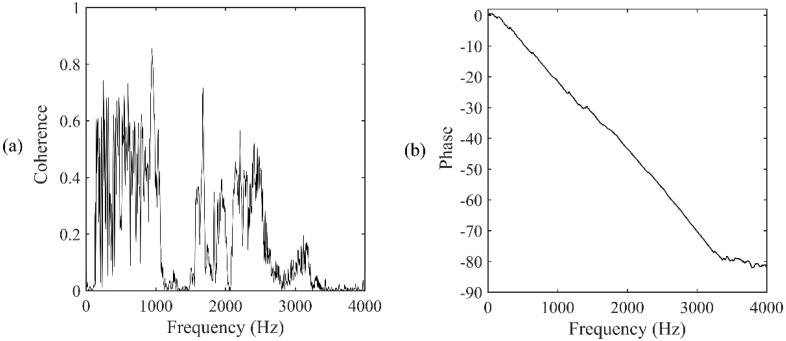
Frequency analysis of the signals measured at A2&A3: (**a**) Coherence function; (**b**) phase spectrum.

**Figure 14 sensors-19-00310-f014:**
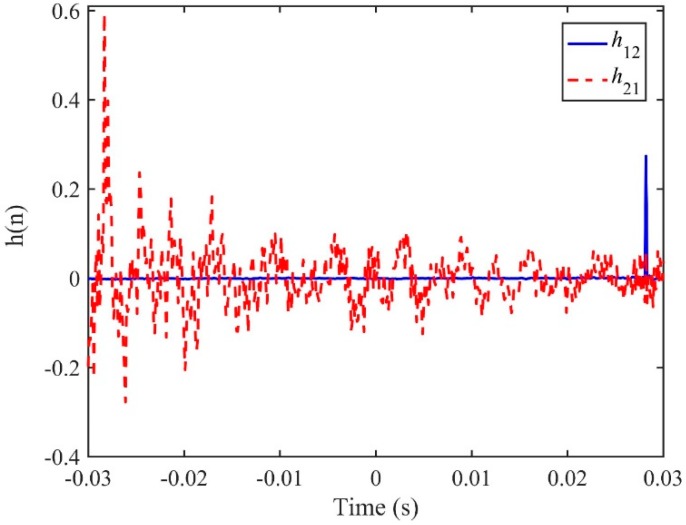
IRFs between the signals measured at A2&A3 using the conventional LMS adaptive method.

**Figure 15 sensors-19-00310-f015:**
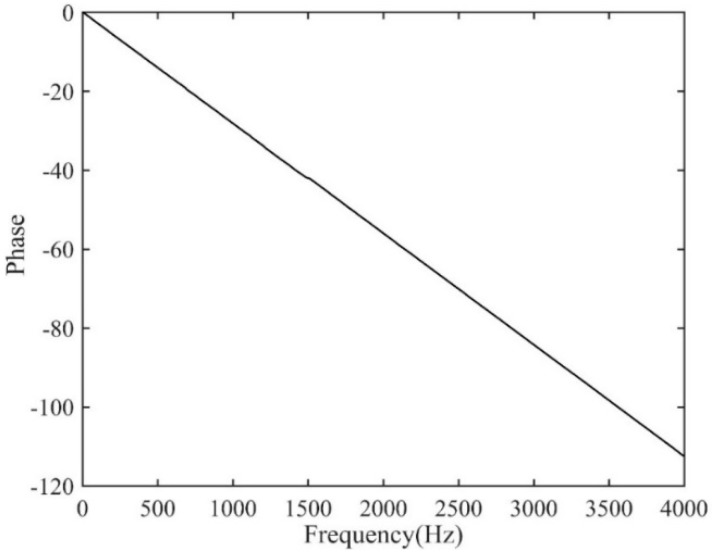
The new FRF between the signals measured at A2&A3.

**Figure 16 sensors-19-00310-f016:**
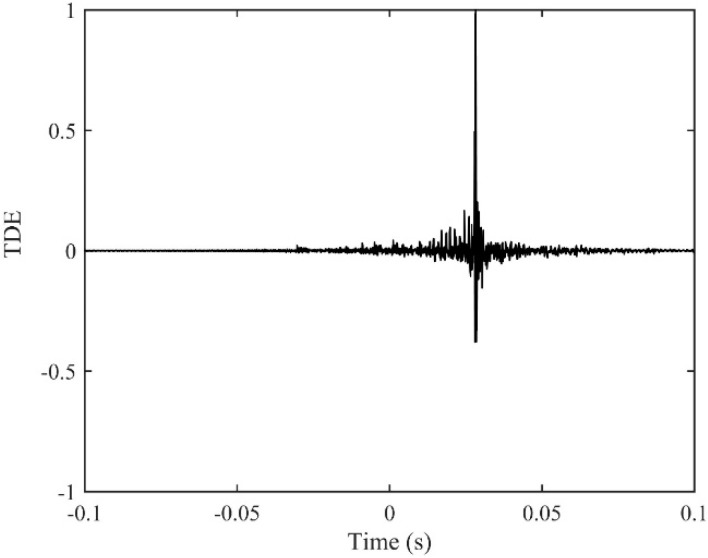
The proposed ADPHAT algorithm for TDE when *D* = 50 m.

**Figure 17 sensors-19-00310-f017:**
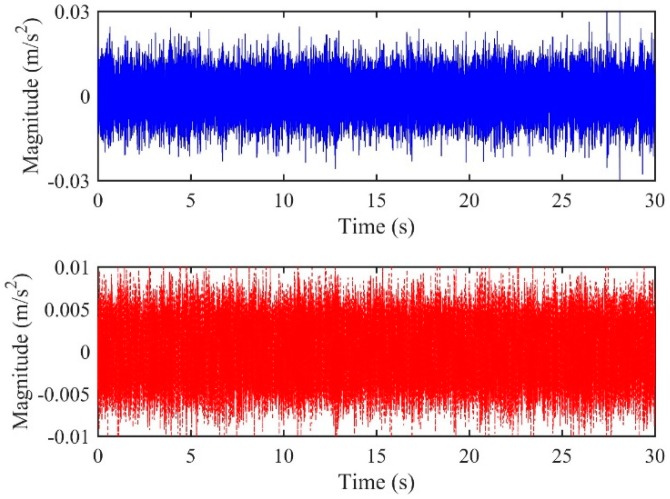
Leak signals measured by the accelerometer pair of (**a**) A1; (**b**) A3.

**Figure 18 sensors-19-00310-f018:**
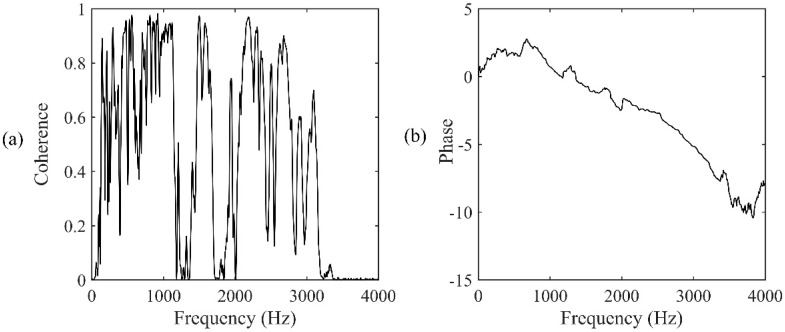
Frequency analysis of the signals measured at A1&A3: (**a**) Coherence function; (**b**) phase spectrum.

**Figure 19 sensors-19-00310-f019:**
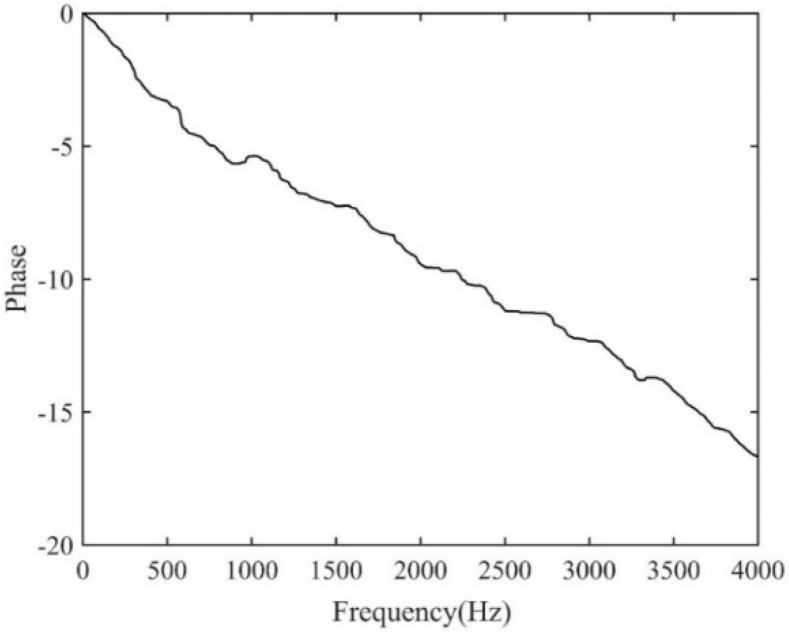
The new FRF between the signals measured at A1&A3.

**Figure 20 sensors-19-00310-f020:**
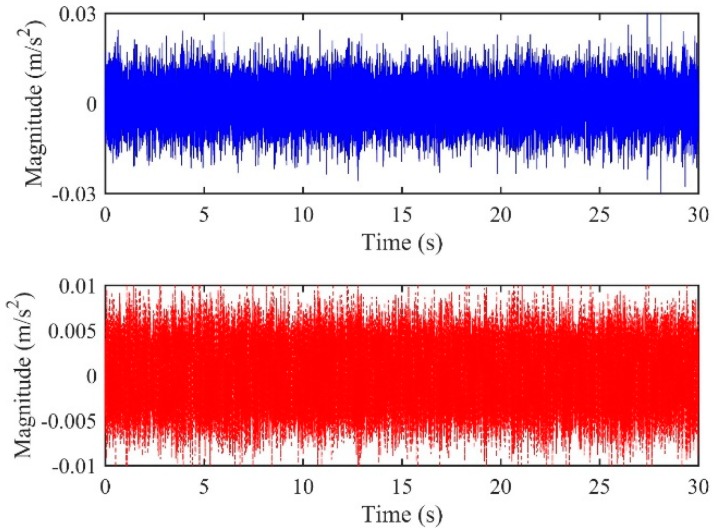
The proposed ADPHAT algorithm for TDE when *D* = 78.8 m.

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
