# Peer review of "Adaptive Phase Transform Method for Pipeline Leakage Detection"

_sensors, 2019, doi:10.3390/s19020310_

Reviewer 1 Report

The paper quality is good. All the sections are correctly described and well written.

I only have 2 minor suggestions to the authors:

in the introduction, avoid using the word "our", and be more generic.

while reporting equation, do not use "sqrt" or "exp", but use their correct simbols.

Reviewer 2 Report

The paper "Adaptive phase transform method for pipeline leakage detection" describes an interesting adaptive phase transform method based on least mean square (LMS) algorithms for the determination of the leak location. Quality of language is good however there are some parts where grammar should be improved.

In the paper you describe that the leak noise is concetrated at low frequencies. It would be sufficient to write frequency range. 

Lines 96-98: text describes the same which is mentioned above in the nomenclature.

Lines 209, 210, 215,... keep units and numbers together - on a same line

Line 212: must be kHz

Figures and figure titles have to be on a same page (e.g. Fig. 5 or 7).

Line 248: it is not clear what authors mean by 2% ~3%...please rewrite to make sentence clear.

Line 250: mdash should be used for minus sign.

Reviewer 3 Report

Authors have presented a detailed approach to detect pipeline leakage using adaptive phase transformation. Manuscript is well written and presents the approach clearly. There are few grammatical corrections. Other than the manuscript is a good read for researchers in the field. 
